# Wear of Ni-Based Superalloy Tools in Friction Stir Processing of Commercially Pure Titanium

**Alihan Amirov \*, Alexander Eliseev \***  **and Vladimir Beloborodov \***

Institute of Strength Physics and Materials Science, Siberian Branch Russian Academy of Sciences, 2/4, pr. Akademicheskii, Tomsk 634055, Russia

\* Correspondence: amirov@ispms.ru (A.A.); alan@ispms.ru (A.E.); vabel@ispms.ru (V.B.)

**Abstract:** Conventional methods for joining titanium alloys often provide a relatively low quality of joints impaired by high residual stresses. A possible solution to this problem can be offered by friction stir welding, which has been increasingly used for aluminum alloys. However, the friction stir welding of titanium alloys is complicated by severe tool wear due to high loads and temperatures in the process. Good results were reported for a tool made of ZhS6U superalloy, but tool life still needs to be improved. Here, we study the wear of a tool made of ZhS32 Ni-based superalloy, which has higher heat resistance than ZhS6U, and the wear of a liquid-cooled ZhS6U tool in the friction stir processing of commercially pure titanium. The effect of tool wear on the strength characteristics of the processed material is discussed. The total processing path length traversed by the tools without failure was 2790 mm. In both cases, the most severe wear was observed at the pin root. Liquid cooling significantly reduced the wear. Based on the obtained results, tool wear is proposed to occur by an adhesion–diffusion mechanism during friction stir processing.

**Keywords:** friction stir welding; commercially pure titanium; Ni-based superalloys; liquid cooling; microstructure; welding tool; tool wear

## 1. Introduction

Friction stir welding (FSW) emerged in the early 1990s as a method for the solid-state joining of various materials [1,2]. It has been widely and successfully used to join aluminum alloys for aerospace applications [3,4]. Another essential material in the aerospace industry is titanium alloys, showing high specific strength, heat resistance, and corrosion resistance [5]. Despite the fact that these alloys have good weldability [6], conventional fusion welding of titanium alloys leads to some undesirable effects, such as the formation of a coarse-grained cast structure, porosity, and residual stresses [7]. That is why fusion-welded titanium parts are often subjected to additional heat treatment, which is often time-consuming and costly due to the large dimensions of welded structures. Friction stir welding shows more promise for joining titanium alloys because some FSWed titanium alloys do not require post-heat treatment.

One of the biggest problems in the FSW of titanium alloys is the rapid tool wear caused by the influence of high temperatures under high loads [8,9]. For these reasons, FSW tools for the welding of titanium alloys are usually made of refractory or cermet/metal matrix materials, including tungsten-based alloys [10–20] such as tungsten carbide or cobalt-based tungsten–rhenium alloys [21–24], and alloys based on molybdenum [25–27], cobalt [28], or polycrystalline cubic boron nitride [29,30]. The major drawbacks of many of these alloys are rapid wear, high cost, manufacturing complexity, and chemical interaction with titanium alloys.

Nickel-based superalloys can be good candidates as FSW tool materials. These alloys have been developed since the middle of the 20th century [31], and there are now five generations of Ni-based alloys that differ in composition, heat resistance, and cost. The

first-generation superalloys contain tungsten, chromium, molybdenum, and niobium and are relatively inexpensive. The most popular alloy of this generation is ZhS6U (analogous to MAR-M247 alloy), which was used as a tool material for the FSW of steel [32] and titanium alloys [33]. Previously, we studied the durability of such a tool and the effect of its wear on the strength of FSWed titanium alloy [34]. The result was satisfactory because the ZhS6U tool was cheaper than, e.g., a tool made of a tungsten–rhenium alloy. The total length of the high-quality joints processed by the tool was more than 1600 mm, which was greater than for a tungsten carbide tool. The second- and third-generation alloys contain rhenium, and the fourth- and fifth-generation alloys contain rhenium and ruthenium. These alloying elements increase the heat-resistant characteristics but make the superalloys expensive.

The purpose of this work was to study methods for increasing the life of nickel-based superalloy tools for the friction stir processing of commercially pure titanium. The aim of the work was also to investigate the effect of tool wear on the structure and properties of the welds. We investigated the wear behavior of a tool made of second-generation ZhS32 Ni-based superalloy and the effect of its wear on the quality of the processed material. It should be noted that the use of tools made of third-, fourth-, and fifth-generation superalloys is not so cost-effective because of the high content of rhenium or ruthenium. In addition, the effect of liquid cooling during FSW on tool life was considered.

## 2. Materials and Methods

### 2.1. Materials and Experimental Set-Up

The tool service life was studied during the friction stir processing (FSP) of solid titanium workpieces under constant process parameters. Friction stir welding and friction stir processing occur by the same mechanism [8], but the FSP technique is easier to perform as there is no need to prepare two workpiece sheets and fix them rigidly. Prior to FSP, titanium workpieces were clamped in an FSW setup (ISPMS SB RAS, Tomsk, Russia). The rotating tool was plunged into the titanium sheet and moved along the workpiece. No pre-drilling was performed. As a result of friction between the tool and the workpiece, the workpiece material was heated to a plastic state and underwent adhesive mass transfer. The dwell time was 1 s. The support plate for the processed titanium sheet was an AISI 304 stainless steel sheet (Figure 1a). The tilt angle of the tool was 1.5°. The processing was carried out to a depth of 2 mm.

The geometry of the FSP tools used is shown in Figure 1b. The material of the first tool was a rhenium-containing second-generation ZhS32 Ni-based superalloy, whose composition is given in Table 1.

**Table 1.** Elemental composition of ZhS32 superalloy (wt%).

| Fe | Nb | Re | Cr | Co | W | Ni | Al | Mo | S |
|---|---|---|---|---|---|---|---|---|---|
| ≤0.5 | 1.4–1.8 | 3.6–4.3 | 4.5–5.3 | 9–9.5 | 8.1–8.9 | 56.88–62.98 | 5.7–6.2 | 0.9–1.3 | ≤0.005 |
| **Ce** | **Si** | **Mn** | **P** | **C** | **Ta** | **Cu** | **B** | **V** | **Y** |
| ≤0.025 | ≤0.2 | ≤0.3 | ≤0.01 | 0.12–0.17 | 3.7–4.4 | ≤0.03 | ≤0.02 | ≤0.15 | ≤0.005 |

The second tool was made of ZhS6U Ni-based superalloy, with the composition presented in Table 2. A similar tool was used in our previous study [34], but here, it was subjected to liquid cooling during FSP. Both tools were made of cast materials and machined by turning from solid cylinders.

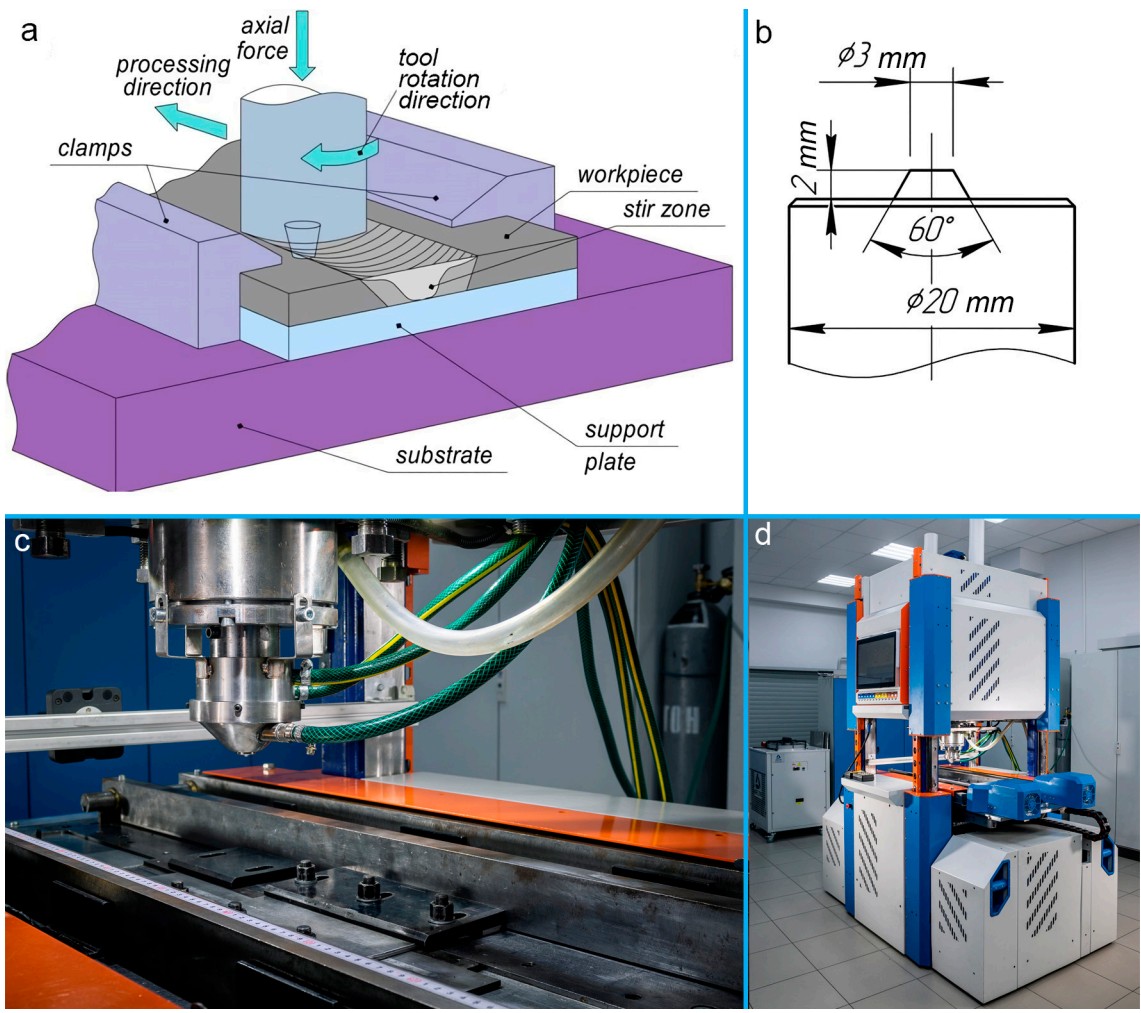

**Figure 1.** Scheme of friction stir processing (**a**), drawing of the tool profile (**b**), appearance of the cooling system (**c**), and experimental setup (**d**).

**Table 2.** Elemental composition of ZhS6U superalloy (wt%).

| Fe | Nb | Ti | Cr | Co | W | Ni | Al | Mo | S |
|---|---|---|---|---|---|---|---|---|---|
| ≤1 | 0.8–1.2 | 2–2.9 | 8–9.5 | 9–10.5 | 9.5–11 | 54.3–62.7 | 5.1–6 | 1.2–2.4 | ≤0.01 |
| **Ce** | **Si** | **Mn** | **P** | **C** | **Zr** | **Bi** | **B** | **Pb** | **Y** |
| ≤0.02 | ≤0.4 | ≤0.4 | ≤0.015 | 0.13–0.2 | ≤0.04 | ≤0.0005 | ≤0.035 | ≤0.01 | ≤0.01 |

The processed workpiece was made of single-phase commercially pure titanium and had the shape of a 2.5-mm thick plate. The initial composition of the titanium alloy is shown in Table 3. This material was chosen because it is easy and convenient to investigate tool wear and the effect of wear on joint strength. In more advanced and complex alloys, phase transitions can occur during the FSP process, which can disturb the experiment's purity. Also, this material was used as a model alloy in previous work, so it is possible to compare the results.

**Table 3.** Elemental composition of titanium alloy (wt%).

| Fe | C | Si | N | Ti | O | H | Impurity |
|---|---|---|---|---|---|---|---|
| ≤0.25 | ≤0.07 | ≤0.1 | ≤0.04 | 99.24–99.7 | ≤0.2 | ≤0.01 | balance 0.3 |

The tool was cooled using a specially designed coolant supply system, as shown in Figure 2. The fluid was injected through the upper nozzle to the cooling system where it flew through the upper channels of the holder to the central pipe connected with the tool and went down to cool the tool. Then, the fluid moved upward through the vertical channels of the sleeve to the lower channels of the holder and flew out through the lower nozzle. Water with a temperature of 18 °C was used as a coolant. At the outlet of the cooling system, the flow rate was measured, which was 2.5 L/min. Based on this flow rate and design, the fluid velocity at the inner wall of the tool was 1.3 m/s. The water flow was adjusted so that there was no leakage between the collar of the cooling system and the spindle. In this way, all of the fluid entering the system flowed out through the outlet nozzle, that is, it flowed through the thin pipe. A protective argon atmosphere was used to avoid titanium oxidation during FSP. Argon was fed into the process zone through the nozzle marked with a green arrow in Figure 2a.

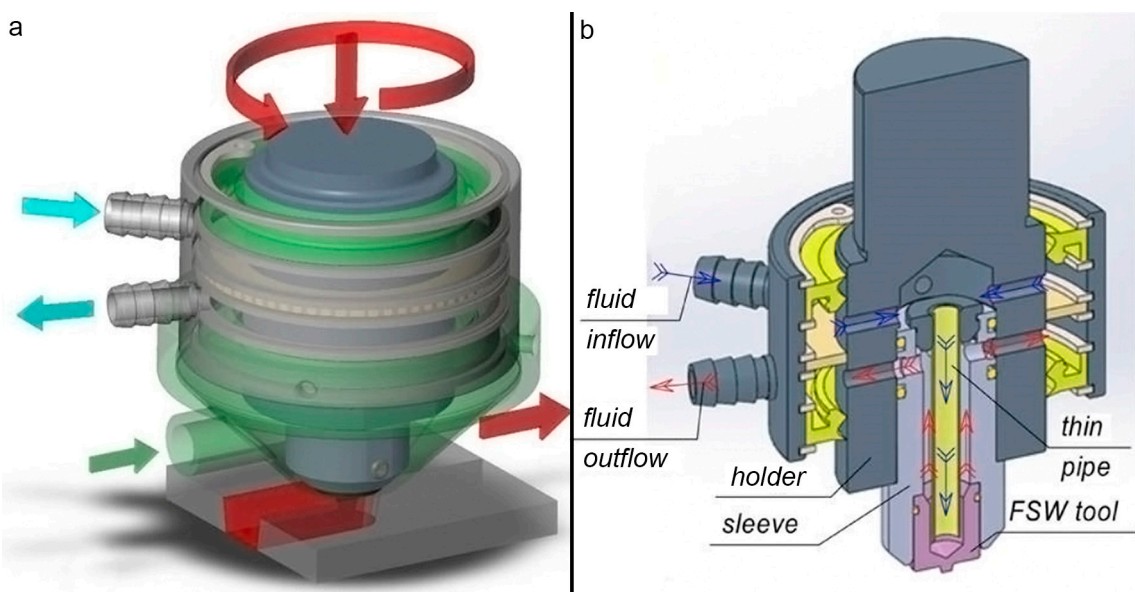

**Figure 2.** Three-dimensional view of the tool cooling system (**a**) and schematic of the tool cooling process during FSP (**b**).

FSP was performed using the process parameters that provided the highest weld strength in our previous studies [33,34]. The axial load on the tool during processing was 950 kgf, the traverse speed of the tool was 180 mm/min, and the rotational speed was 950 rpm. The total path length traveled by both tools in titanium alloy with the given process parameters was 2790 mm. In [34], processing with a ZhS6U alloy tool without cooling was carried out in 100 mm length passes. Here, we used the same pass length for the correct comparison of tool wear behavior.

### 2.2. Investigative Techniques

Examinations were carried out on both the processed material and the tool material. Tool samples for metallographic examination were cut by electrical discharge machining along the axial direction of the tool using a DK7750 EDM machine (Suzhou Simos CNC Technology Co., Ltd., Suzhou, China). Cutting was performed using molybdenum wire

with a diameter of 0.1 mm. Metallographic studies of the workpiece material were carried out on cross-sectional samples cut perpendicularly to the processing direction. The samples were polished with abrasive papers to 2000 grit and diamond paste (1/0 grit). Then the tool samples were etched to reveal the microstructure with a solution of composition 8 g $CuSO_4$ + 40 mL HCl + 40 mL $H_2O$. The structure of the workpiece material was revealed by keeping the samples in a 2% aqueous solution of HF for two minutes and then washing them in a 40% aqueous solution of $HNO_3$.

Metallographic microstructural analysis of the tool and workpiece materials was performed on an Altami MET-1C optical microscope (Altami, St. Petersburg, Russia) and an Apreo S LoVac scanning electron microscope (Thermo Fisher Scientific, Portland, OR, USA). The tool was photographed after each pass during processing to observe its shape change. Adobe Photoshop (Version 23.5) and ImageJ 1.54d software were used to process the digital images.

In order to evaluate the effect of tool wear on the strength characteristics of the processed material, the workpiece samples (hereinafter referred to as the weld samples for convenience) were subjected to quasi-static tensile tests with stretching across the weld. Mechanical tests were conducted on a UTC 110M-100 testing machine (Test Systems, Ivanovo, Russia) at room temperature with a stretching rate of 1 mm/min. The cross-sectional microhardness of the tools after processing was measured by the Vickers method. Microhardness measurements were made on polished (not etched) samples. To establish the possible effect of friction on the structure of the tool surface layers, measurements were performed at different depths from the friction surface down to the bulk. The measurements were carried out at a 50 g load and a 10 s dwell time. This load was chosen to reduce the size of the indentation so that the step was as small as possible and measurements could be taken as close to the edge of the sample as possible. The measurement step was varied from 0.025 to 0.1 mm. The first indent in the series was made at a distance of 0.017–0.02 mm from the friction surface. The distance between the indentations was not less than 2 diagonals. For this purpose, measurements near the friction surface were made in staggered patterns. The microhardness of the samples was measured using a Duramin 5 microhardness tester (Struers, Ballerup, Denmark).

## 3. Results and Discussion

### 3.1. Wear and Structure of the ZhS32 Ni-Based Superalloy Tool

Here, we compare the wear behavior of a ZhS32 tool with the previously investigated wear behavior of a ZhS6U tool [34]. The general view of the ZhS32 tool after a different number of passes is shown in Figure 3. A visual inspection of the tools showed that there were residues of adhered titanium on the friction surfaces. The photographic images show that the surface had an oxidizing color, indicating the presence of an oxide film. As in [34], the most severe wear during FSP occurred at the root of the pin. The greatest wear in the pin area may be due to the fact that this is the area where the maximum welding temperature was observed, as shown in the paper [35]. The wear at the edge of the pin tip was less pronounced. The pin geometry during wear gradually changed from tapered cylindrical to straight cylindrical. The tapered geometry was preserved after traversing a distance of 1140 mm, which is 35 mm larger than in the ZhS6U tool [34]. This difference in tool wear is insignificant for industrial applications.

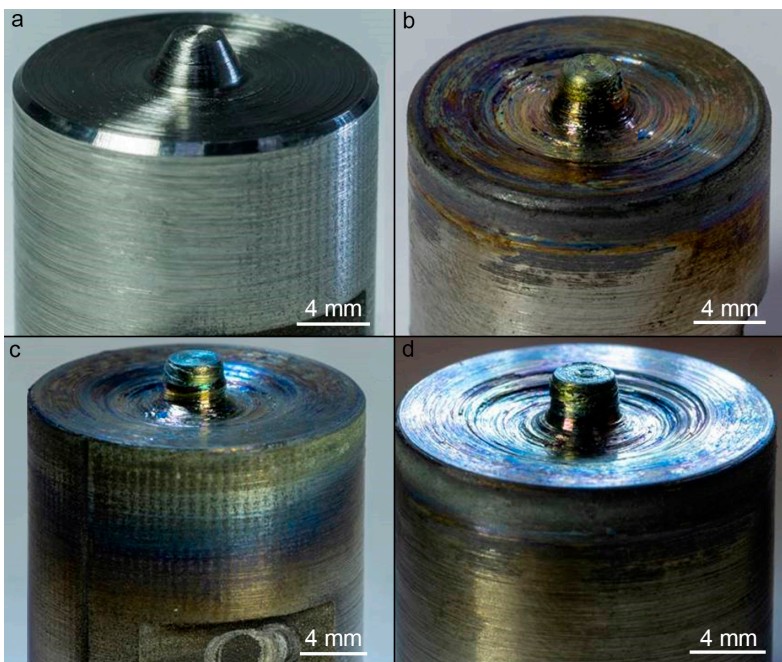

**Figure 3.** General view of the ZhS32 tool before friction stir processing (**a**) and after traversing a distance of 1140 mm (**b**), 1980 mm (**c**), and 2790 mm (**d**).

Figure 4 presents the axial cross-sectional view of the tool at a distance of 2790 mm from the weld start, with a superimposed drawing of the initial tool geometry. The maximum wear is observed at the pin root, where the tool was worn to a depth of 1 mm. The pin wear was less than that in [34], because the pin root diameter after FSP in this work was larger by 0.3 mm and was equal to 3.6 mm. The shoulders closer to the edge were worn to a depth of about 0.35 mm. The height of the pin did not change. This type of wear can cause some problems. With a smaller pin root diameter, there may be insufficient heat generation due to lower friction velocity and insufficient mixing. In addition, if the shoulders are worn more severely than the pin, the tool is plunged deeper into the material and the pin can come into contact with the substrate. As a result, the workpiece can be welded to the substrate and the substrate material can be stirred into the stir zone.

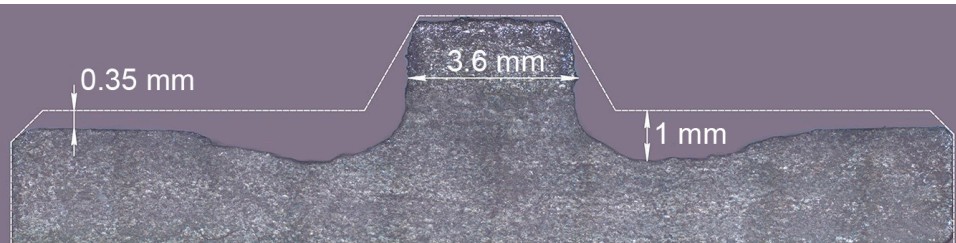

**Figure 4.** Axial cross-sectional view of the tool after traversing a distance of 2790 mm with a superimposed drawing of the initial tool geometry.

The metallographic image of the tool structure exhibits γ-phase dendritic colonies (Figure 5). The appearance of the colonies in the worn tool differed significantly from that in the as-cast ZhS32 billet (Figure 6a). The shapes of the dendritic colonies in the tool were predominantly elongated, while in the as-cast metal, they were nearly equiaxed. This was due to the cast structure of the billet, in which dendrites grew away from the surface. The tool and the as-cast billet were cut at different angles for structural examination. The cutting was carried out at a random angle because it is impossible to know the direction of the dendrites before cutting. However, this did not interfere with the measurement of the dendritic colony sizes and primary and secondary arm spacings. The arm spacing did

not change when the colony rotated, because it was still the same colony. Because of the change in perspective, there may have been a large error in the measurements due to the viewpoint.

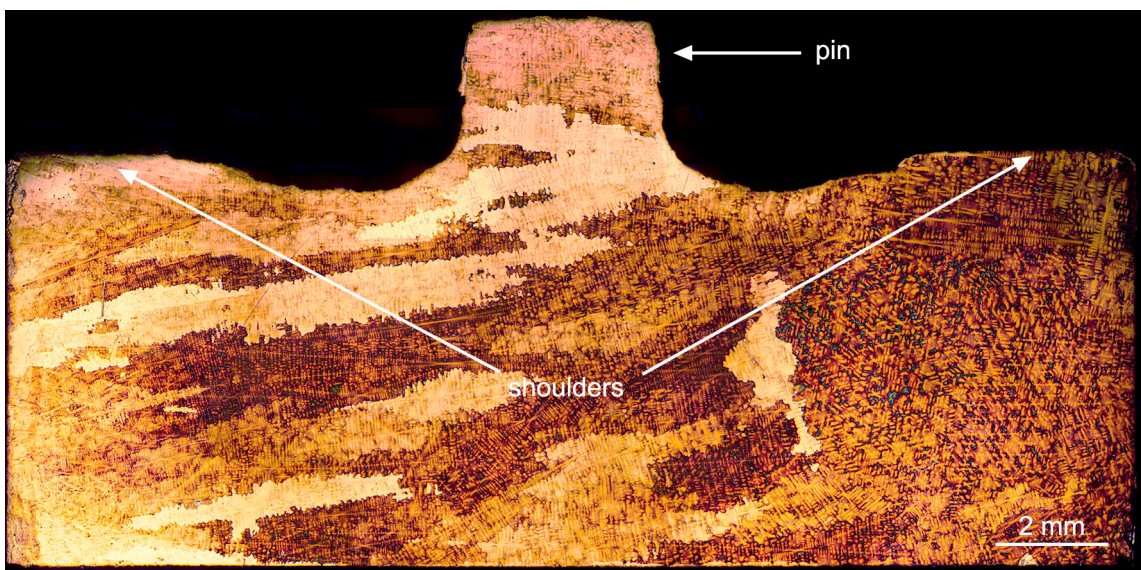

**Figure 5.** Axial metallographic section of the ZhS32 tool after traversing a distance of 2790 mm.

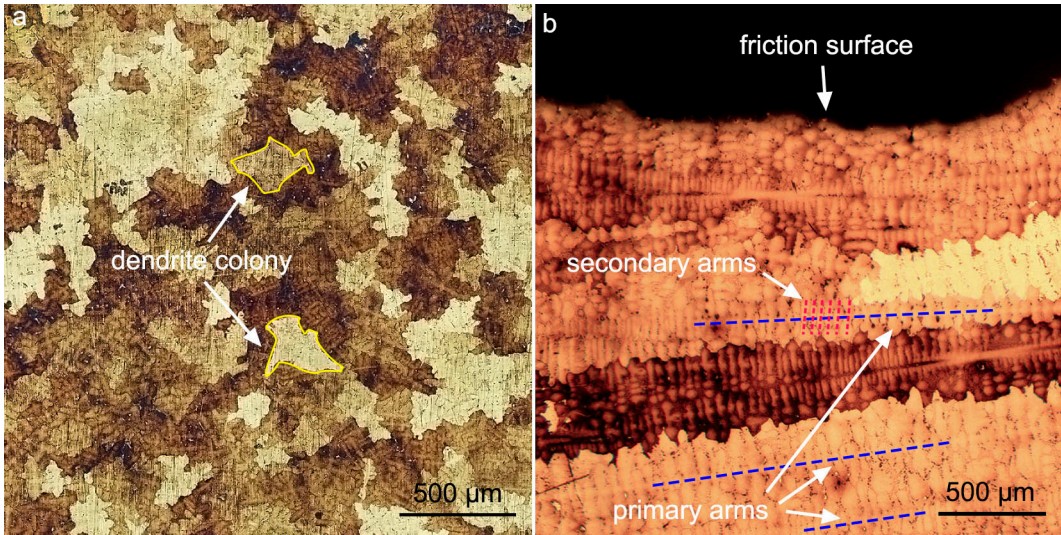

**Figure 6.** Metallographic sections of as-cast ZhS32 alloy (**a**), and the alloy structure near the friction surface of the tool after FSP (**b**).

Figure 6b shows a metallographic section of the tool in the most worn region at the pin root. Comparison of the structure near the friction surface and in the bulk indicates that the friction during processing had no specific effect on the dendritic structure of the surface layer at this hierarchical level. In particular, there were no differences in the image contrast and no differences in the dendrite size at the friction surface and in the amount of the instrument. However, high temperatures during the FSP led to a significant change in the structure of the tool as a whole. The size of the dendritic colonies increased by more than a factor of 3.5, and the primary and secondary arm spacings increased by a factor of about 1.5. The measurement results are presented in Table 4. The microstructural parameters were measured using a ruler on metallographic images. At least 20 measurements were made to obtain the standard deviation for each parameter.

**Table 4.** Microstructural parameters of as-cast and ZhS32 alloy and tool material after FSP.

| Parameters | Base Material | Tool Material |
|---|---|---|
| Size of dendritic colonies, mm | $0.36 \pm 0.09$ | $1.36 \pm 0.66$ |
| Primary arm spacing, μm | $231 \pm 33$ | $316 \pm 80$ |
| Secondary arm spacing, μm | $26 \pm 3$ | $39 \pm 5$ |

　　　The cross-sectional microhardness measurement results for the ZhS32 tool after FSP and the measurement diagram are shown in Figure 7. The measurements were performed in the regions of maximum tool wear and maximum/minimum friction velocities. The results show that the microhardness of the entire tool was 10% lower than that of the as-cast ZhS32 alloy and differed slightly in different regions. The microhardness near the friction surface did not change. Large individual deviations could be due to the impact of the indenter on carbides present in the alloy. These results are in good agreement with the microstructural observations. In particular, the structure of the tool material coarsened, which could lead to a lower microhardness. The metallographic data indicate that the structure changed evenly throughout the tool, so there was no difference between the microhardness values near the friction surface and in the bulk. Some similar results were observed in [36], where steel tools were investigated after the FSW of aluminum alloy 6061. In particular, it was found that heat effects during the welding led to a decrease in the micro-hardness of the tool. This was attributed to structure coarsening, annealing, and stress release. The largest decrease in hardness was found in the tool pin.

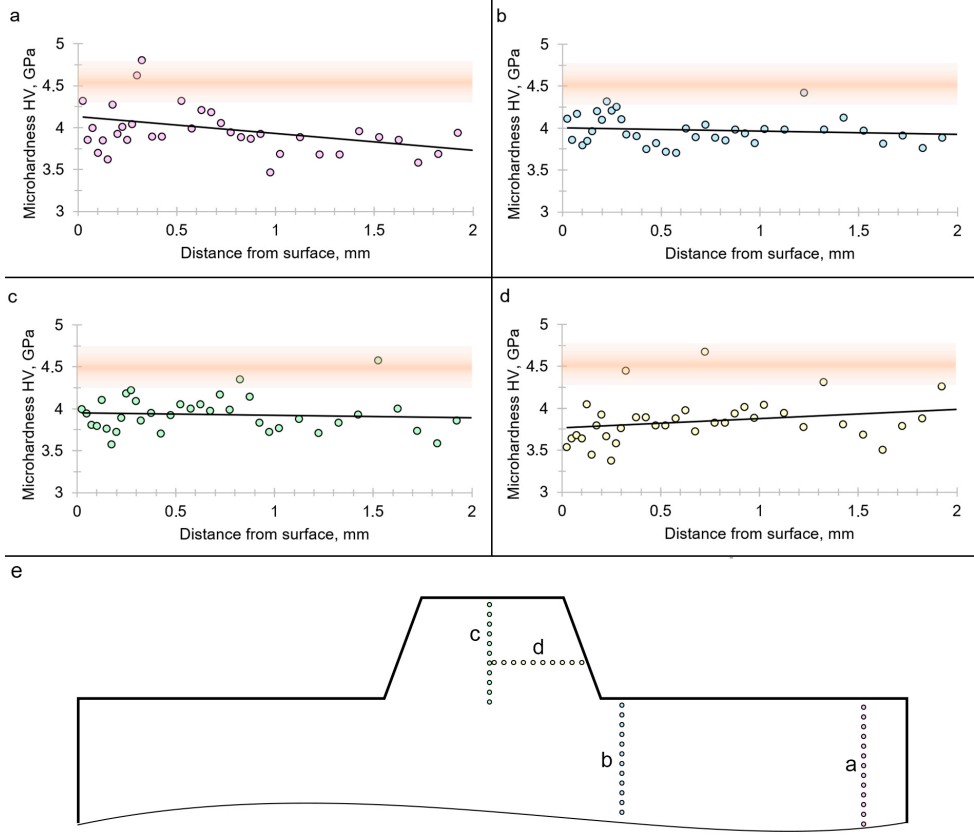

**Figure 7.** Microhardness measurement results for the ZhS32 tool after FSP measured along different directions indicated in the diagram (**e**): (a) A direction, (b) B direction, (c) C direction, (d) D direction. The black solid lines in (**a–d**) show the linear approximation of the measured data. The light orange line indicates the microhardness of the as-cast ZhS32 alloy with an allowance for error.

### 3.2. Quality of Friction Stir Processing by the ZhS32 Tool

The quality of the friction stir processing was evaluated by the structure and mechanical properties of the processed material. A cross-sectional metallographic examination of the workpiece material revealed the same appearance of the weld and weld zones along the centerline in all samples (Figures 8–10). The samples shown in the figures were cut at distances of 5, 240, and 2790 mm from the start of the weld, respectively. There are two typical weld zones: the stir zone (SZ) in the weld center, which is in direct contact with the tool during processing and consists of fragmented and recrystallized titanium alloy grains, and the thermomechanically affected zone (TMAZ) at the stir zone edges, which is not in contact with the tool but is affected by the temperature and deformation. It should be noted that the TMAZ formed in FSPed titanium alloys is narrower than in other more ductile materials. In general, the geometry of the weld zones is symmetrical with respect to the weld axis. The average width of the TMAZ is about 0.25 mm. The heat-affected zone, which is usually formed between the base metal (BM) and the TMAZ in FSW joints or FSP material, was not observed metallographically, which is typical for commercially pure titanium. The shapes and dimensions of the weld zones in the material processed by different tools were generally similar; the TMAZ thickness differed slightly. Metallographic sections were taken approximately every 250 mm. The first sample cut at a distance of 5 mm from the start of the weld had no defects, as shown in Figure 8. Its appearance was typical for friction stir processing. The stir zone area was $11.47 \pm 0.05$ mm$^2$.

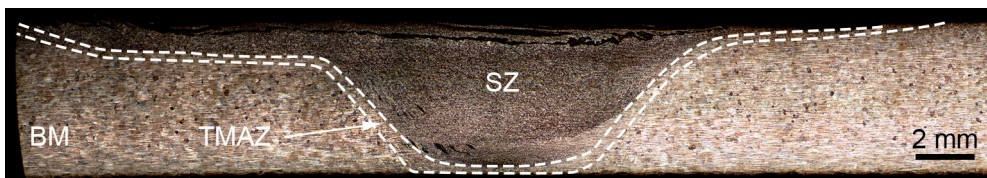

**Figure 8.** Cross-sectional view of the FSPed titanium sample at a distance of 5 mm from the start of the weld obtained using the ZhS32 tool.

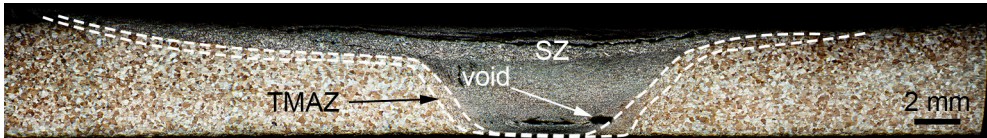

**Figure 9.** Cross-sectional view of the FSPed titanium sample at a distance of 240 mm from the start of the weld obtained using the ZhS32 tool.

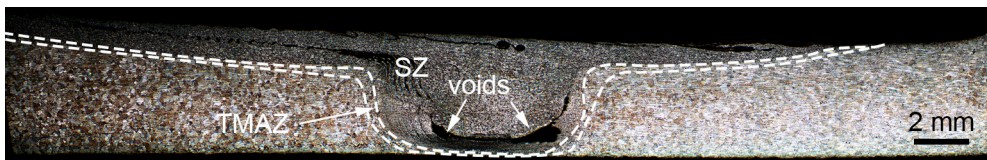

**Figure 10.** Cross-sectional view of the FSPed titanium sample at a distance of 2790 mm from the start of the weld obtained using the ZhS32 tool.

The first macrodefects, in the form of voids, were observed in the sample cut at a distance of 240 mm, in the lower part of the stir zone (Figure 9). The cross-sectional area of the stir zone in this sample was slightly smaller and equal to $11.41 \pm 0.05$ mm$^2$.

These defects became larger and new defects appeared with increasing weld length. A sharp increase in the volume of the voids was observed at a distance of $\approx 1600$ mm. The last FSPed titanium sample had two large defects (Figure 10). In addition, there was a thick defect layer in the upper part of the stir zone, which could also adversely affect the weld

quality. These objects are similar to those investigated in [37]. This paper shows that these objects were tool wear debris that diffused with the titanium and formed intermetallic compounds of varying composition and stoichiometry. These IMCs were strongly etched during the etching, so they look like voids in the metallographic images. It should be noted that the stir zone width decreased with tool wear in the same way as in the case of using the ZhS6U tool [34]. This is because the pin diameter decreased with tool wear and the volume of the stirred material was reduced, resulting in a narrower stir zone. The cross-sectional area of this zone, as shown in Figure 10, was $6.95 \pm 0.01$ mm$^2$.

As in the case of the defects, the metallographic samples taken from the beginning of the weld were almost free from IMCs. The number of IMCs increased with the processing path length. The appearance of discontinuities and their growth was particularly associated with tool wear, because changes in the tool geometry led to changes in the mass transfer behavior of the processed material. With maintained process parameters, this led to a decrease in heat generation and deformation and, as a result, to poorer stirring. According to the classical heat release equation for FSW [38], the heat input is directly proportional to the linear friction velocity. Decreasing the pin diameter led to decreasing the friction velocity at the same RPM, so the heat generation also decreased. It was also noted in [39] that deviation from the optimal welding parameters led to defect formation. The work [40] experimentally showed that different pin shapes led to different heat generation and different grain structures of the welds. The total cross-sectional area of all weld discontinuities at the final stages of processing was smaller than in our previous study [34], despite the fact that the first discontinuities appeared earlier in the processing. The area of discontinuities was 0.04 mm$^2$ at a distance of 5 mm from the weld start, 0.13 mm$^2$ at 240 mm, and 0.24 mm$^2$ at 2790 mm.

The results of the static tensile tests on workpiece samples are presented in Figure 11. It can be seen that the tensile strength of the samples decreased with tool wear and amounted to 260 MPa at a distance of 2700 mm from the start of processing. Noteworthy is the fracture location in the samples. When testing the workpiece samples taken at distances from 0 to 240 mm, fracture occurred in the base metal. In the distance range from 240 to 1600 mm, fracture occurred in the TMAZ. And in the range from 1600 to 2790 mm, it occurred in the stir zone. A sharp decrease in the strength of the material was observed at a distance of 1600 mm, where the volume of discontinuities increased sharply. Thus, the strength decrease was, first of all, due to the growth of defects with tool wear because the shape change of the pin led to insufficient heating and inadequate stirring of the material. The maximum tensile fracture strain was 19.7% of the base metal, which is equal to 29.5%.

The obtained results and comparison with the data of [34] indicate that the use of ZhS32 or ZhS6U alloy as an FSP tool material made no significant difference in the tool durability or quality of the processed material.

### 3.3. Wear and Structure of the ZhS6U Tool in Friction Stir Processing with Liquid Cooling

The ZhS6U alloy tool was also tested for wear in the friction stir processing of titanium with liquid cooling. The tool profiles at different stages of the experiment are shown in Figure 12. As in the previous section, the friction surface of the tool was covered with a layer of oxidized titanium. It was found that the edges and the root of the pin were the first to wear during processing. The pin shape also gradually changed from tapered to nearly straight cylindrical, but the wear rate was lower than in the ZhS32 tool. The tapered shape of the pin was maintained after traversing a distance of 1950 mm, and the shoulders were slightly concaved closer to the pin.

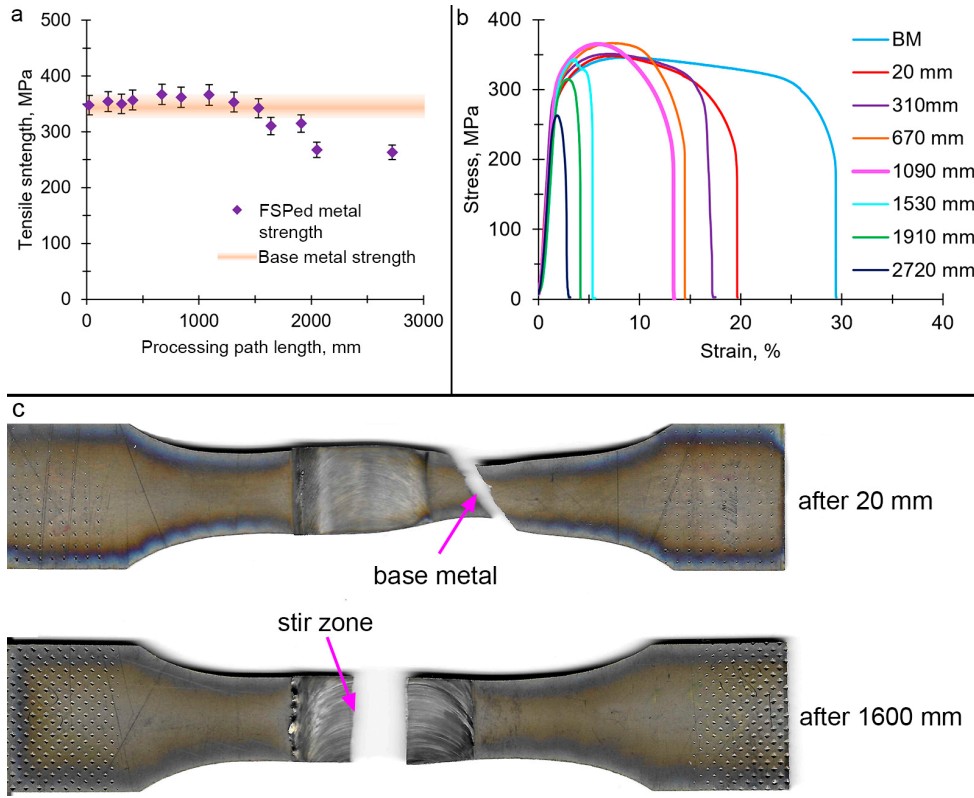

**Figure 11.** Dependence of the weld strength on the tool wear rate (**a**), stress–strain curves of FSPed samples processed by the ZhS32 tool at different processing path lengths (**b**), and image of samples after fracture (**c**).

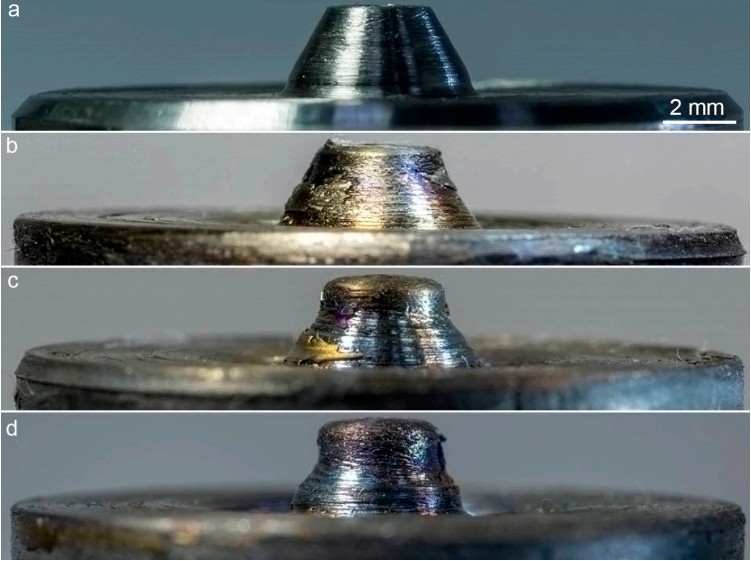

**Figure 12.** ZhS6U tool before friction stir processing with liquid cooling (**a**), and after traversing a distance of 1110 mm (**b**), 1950 mm (**c**), and 2790 mm (**d**).

The axial cross-sectional view of the tool at a distance of 2790 mm from the start of the weld with superimposed drawings of the initial tool geometry and the tool geometry after processing without cooling [34] is shown in Figure 13. As in the case of FSP without liquid cooling, the most severe wear was observed in the shoulders at the pin root. The depth of the depression in this zone was also 1 mm. The wear at the edge of the shoulders

was less pronounced, with an average depth of 0.25 mm. The height of the pin remained unchanged. It should be noted that liquid cooling of the tool led to a lower wear rate of the pin during processing. This was probably due to a decrease in diffusion due to the cooling of the tool. As a result, the pin retained its tapered shape and the root diameter practically did not change, unlike the ZhS32 tool and the ZhS6U tool discussed in [34].

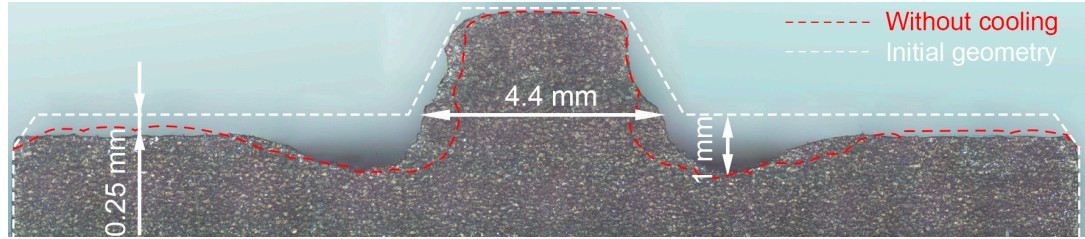

**Figure 13.** Axial cross-sectional view of the ZhS6U tool at a distance of 2790 mm from the start of friction stir processing with liquid cooling, with a superimposed drawing of the initial tool geometry.

Figure 14 shows a cross-sectional metallographic image of the ZhS6U tool after FSP with liquid cooling. The tool structure exhibited $\gamma$-phase dendritic colonies, which trended toward an equiaxed shape after processing, as in the base metal (Figure 15a). However, the dendritic structure coarsened significantly due to the thermal influence of the FSP.

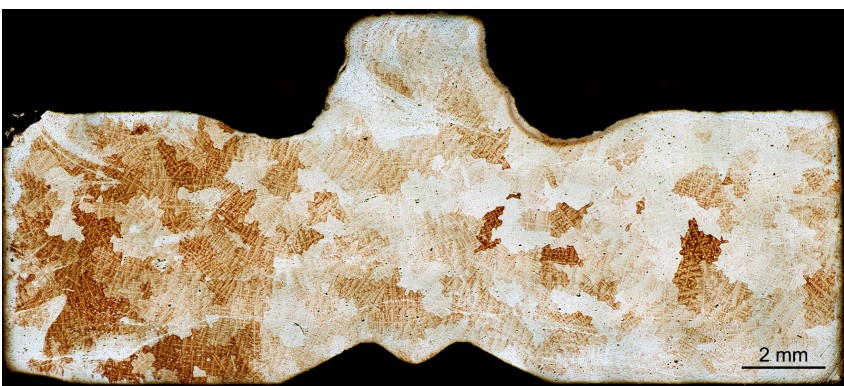

**Figure 14.** Axial cross-sectional metallographic image of the ZhS6U tool at a distance of 2790 mm from the start of friction stir processing with liquid cooling.

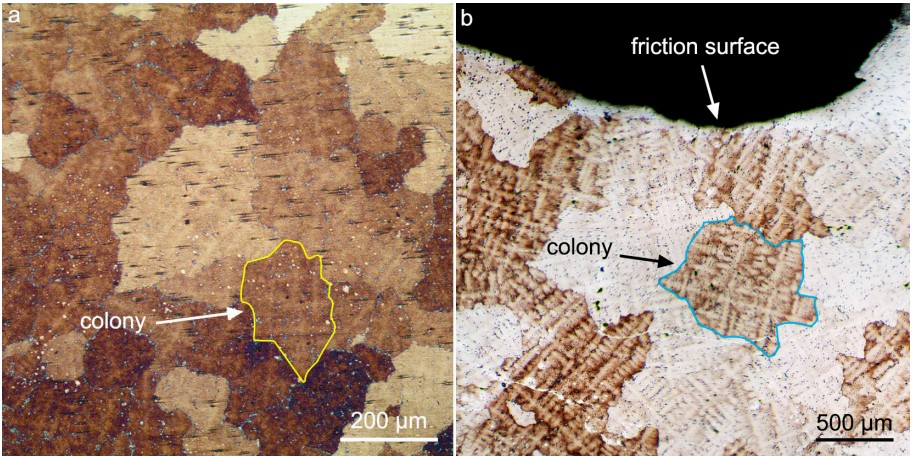

**Figure 15.** Metallographic images of as-cast ZhS6U alloy (**a**), and the alloy structure near the friction surface of the tool after FSP with liquid cooling (**b**).

The dimensions of the structural elements are given in Table 5. As in the ZhS32 tool, the structure changed uniformly throughout the tool. The size of the colonies changed slightly compared to the ZhS6U tool described in [34], where it was equal to $0.8 \pm 0.3$ mm, i.e., by the amount of error. The same can be said about the primary and secondary arm spacings, which were, respectively, $191 \pm 31$ and $44 \pm 6$ μm for the tool without cooling. Compared to the base metal, the size of the dendritic colonies in the tool material increased by a factor of 4.8, and the primary and secondary arm spacings increased by factors of 3.7 and 3.4, respectively. In addition, there was a contrast change in some tool areas near the friction surface, caused by over-etching due to the surface tension at the edge of the sample when it was removed from the reagent. The obtained results indicate that the tool cooling during the FSP did not inhibit the coarsening of the structure. However, supposedly, the lower wear rate was not associated with the dendritic structure evolution but with other mechanisms.

**Table 5.** Microstructural parameters of as-cast ZhS6U alloy and the tool material after FSP with liquid cooling.

| Parameters | Base Material | Tool Material |
|---|---|---|
| Size of dendritic colonies, mm | $0.27 \pm 0.09$ | $1.3 \pm 0.3$ |
| Primary arm spacing, μm | $66 \pm 5$ | $229 \pm 42$ |
| Secondary arm spacing, μm | $14 \pm 1$ | $48 \pm 7$ |

Figure 16 shows a cross-sectional SEM image of the tool in the most worn region. The image was obtained in the phase contrast mode. The dendritic structure of the ZhS6U alloy is clearly seen in the lower part of the image. The light contrast areas are carbides typical of the alloy. The friction surface of the tool was covered with a layer of adhered titanium. The layer was not continuous and varied in thickness. The maximum thickness was about 80 μm. It was found that the titanium layer had a layered structure formed by the repeated deposition of titanium on the tool surface in the course of the adhesive–cohesive interaction during the processing. The adhered titanium and the tool surface were separated by a diffusion layer up to 2.5 μm thick, which had a different contrast from the titanium and ZhS6U alloy.

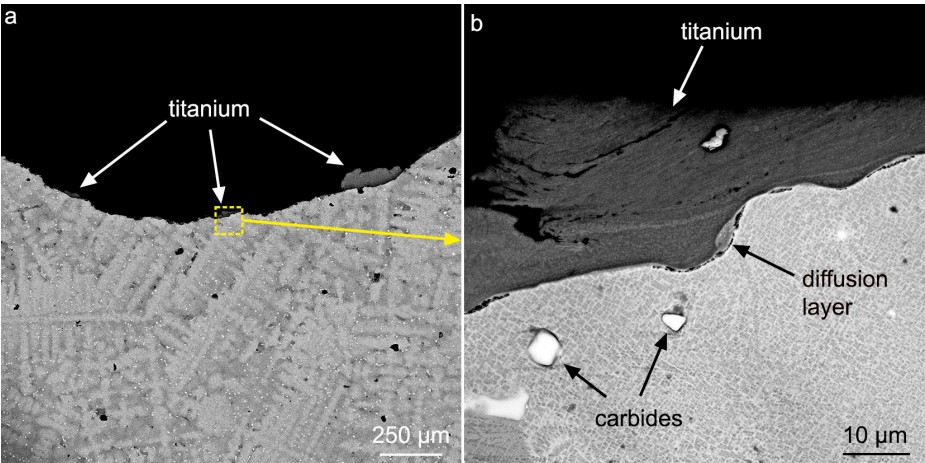

**Figure 16.** Cross-sectional SEM images of the most worn region of the tool (**a**), and a magnified image of the titanium layer adhered to the friction surface (**b**).

EDX analysis of the area around the diffusion layer revealed the presence of titanium and ZhS6U alloy elements, such as W, Ni, Al, and Cr (Figure 17). After the titanium adhered to the tool surface, the elements diffused at a high temperature to form a transition layer.

Noteworthy is that this layer was observed only under the adhered titanium. All other areas of the friction surface, where the titanium separated after the removal of the tool, had no diffusion layer, although the entire tool surface was undoubtedly in contact with the titanium and was subject to wear during the FSP. This may indicate that the images show the moment of diffusion-layer formation. In other regions, where adhered titanium was not observed, the diffusion layer separated together with the titanium.

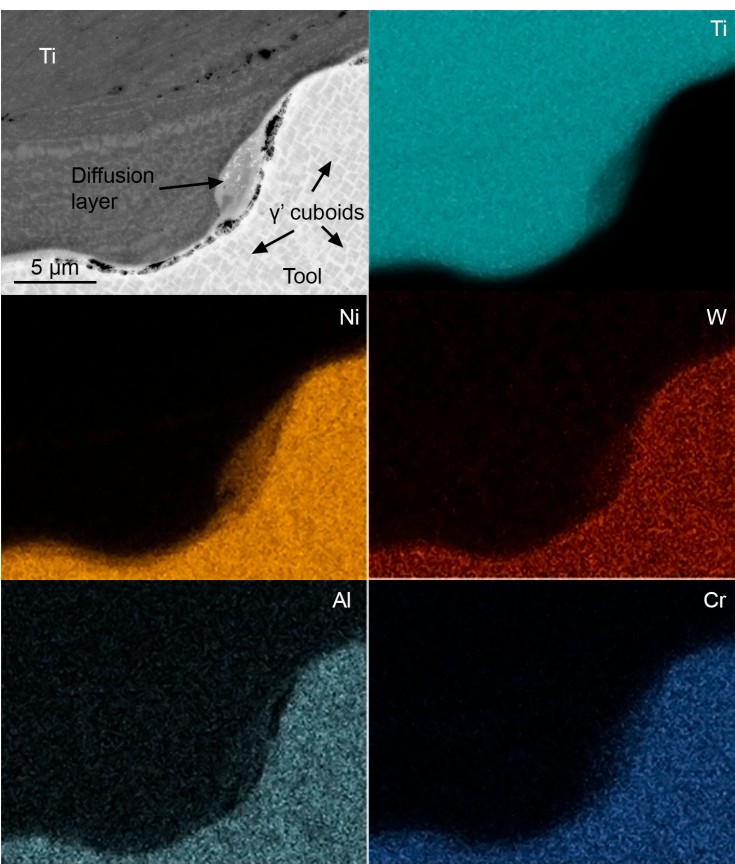

**Figure 17.** EDX map of the tool area around the diffusion layer.

The microhardness measurement results for the ZhS6U tool after FSP with cooling and the measurement diagram are shown in Figure 18. One can see that the tool microhardness did not change compared to the microhardness of the as-cast ZhS6U alloy. The microhardness near the friction surface also remained unchanged. This contradicts the previous results. As in the case of the ZhS32 alloy, the structural characteristics of the ZhS6U increased (Table 5), while the microhardness did not decrease. Presumably, this was due to the presence of residual stresses caused by the cooling of the tool after strong heating during the FSP.

### 3.4. Quality of Friction Stir Processing by the ZhS6U Tool with Liquid Cooling

Figure 19 shows the cross-sectional metallographic images of the welds at distances of 5, 1450, and 2780 mm from the start of processing. A macrostructural examination revealed that two structural zones were formed in all welds: the stir zone and the thermomechanically affected zone. The average thickness of the TMAZ was ≈0.25 mm. As in the previous cases, the heat-affected zone was not identified. The geometry of the structural zones was also symmetrical with respect to the processing axis, and their shapes differed slightly in different welds.

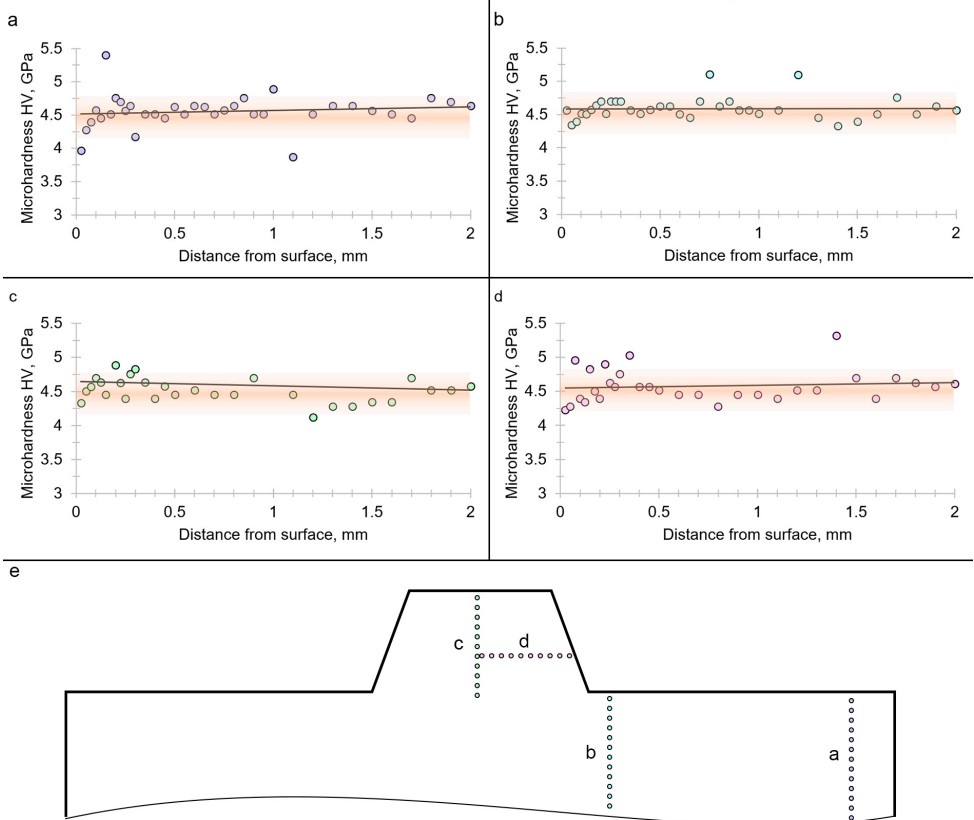

**Figure 18.** Microhardness measurement results for the ZhS6U tool after FSP with cooling measured along different directions indicated in the diagram (**e**): (a) A direction, (b) B direction, (c) C direction, (d) D direction. The black solid lines in (**a–d**) show the linear approximation of the measured data. The light orange line indicates the microhardness of the as-cast ZhS6U alloy with an allowance for error.

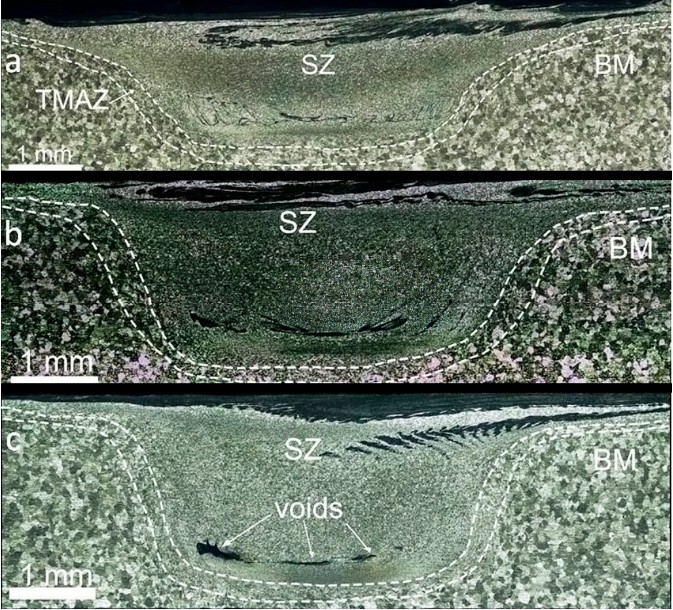

**Figure 19.** Structure of FSPed titanium at distances of 5 (**a**), 1450 (**b**), and 2780 mm (**c**) from the start of processing by the liquid-cooled ZhS6U tool.

As in the previous experiments, the size of the stir zone was also reduced with tool wear, although less drastically. Particularly, the stir zone areas in the first and last samples were 10.23 mm$^2$ and 9.47 mm$^2$, respectively. This is because the pin wear rate was lower in the considered case. The first macrodefects were observed in the cross-section at 1450 mm after the weld start in the lower part of the stir zone. The size and number of defects increased gradually with the weld length. The total area of all discontinuities was equal to 0.04 mm$^2$ after 5 mm of processing, 0.09 mm$^2$ after 1450 mm, and 0.11 mm$^2$ after 2780 mm. The nozzles for argon supply during processing without cooling by the ZhS32 tool and with cooling by the ZhS6U tool are shown in Figure 20a,b. The diameter of the second nozzle was much smaller than in the first case, and therefore, the argon velocity was much higher at the same argon flow rate. The higher argon concentration in the process zone inhibited oxidation.

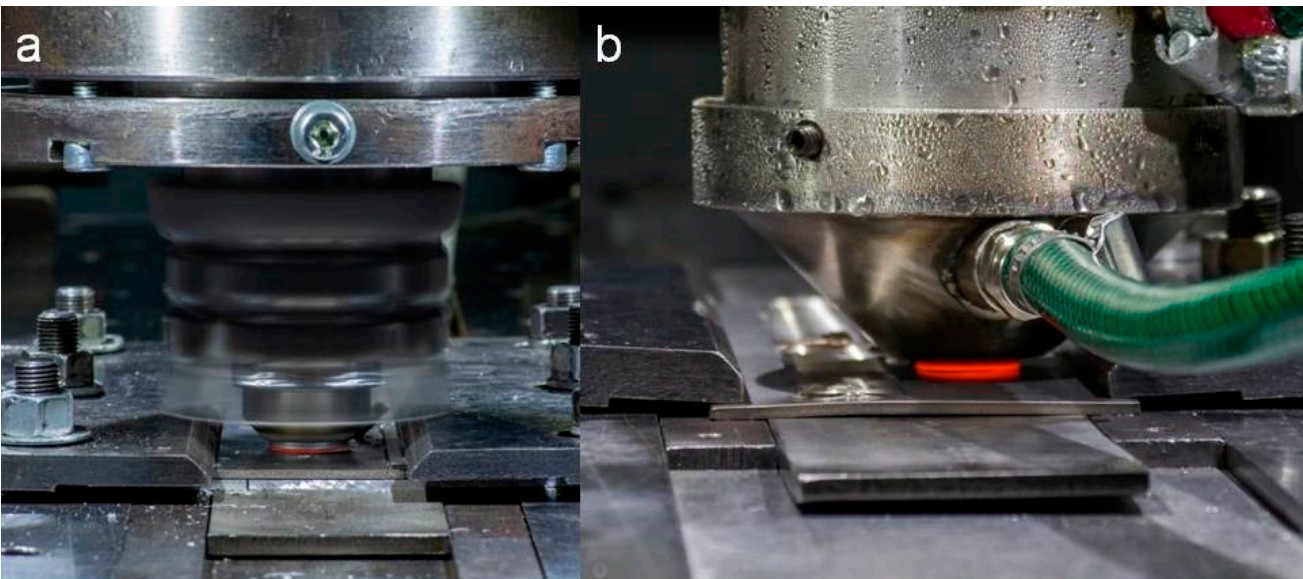

**Figure 20.** Argon inlet nozzle used in FSP experiments without liquid cooling (**a**) and with liquid cooling (**b**).

The results of the quasi-static tensile tests of the weld samples are presented in Figure 21. Specimens for the mechanical tests were cut from the workpiece in the direction perpendicular to the processing axis. The tensile strength of the processed material did not change significantly with tool wear and was, on average, 360 ± 5 MPa (103% of the base material). The relative elongation of the test specimens was almost the same as in the previous experiments. Fracture in all specimens occurred in the base metal or in the TMAZ, except for the last sample cut at a distance of 2780 mm from the weld start, where fracture occurred in the stir zone. This result seems satisfactory, as the weld strength was stable throughout the processing.

Generally, liquid cooling of the tool during the FSP led to positive results, such as a lower tool wear rate and a longer tool life, with preserved quality of the processed material.

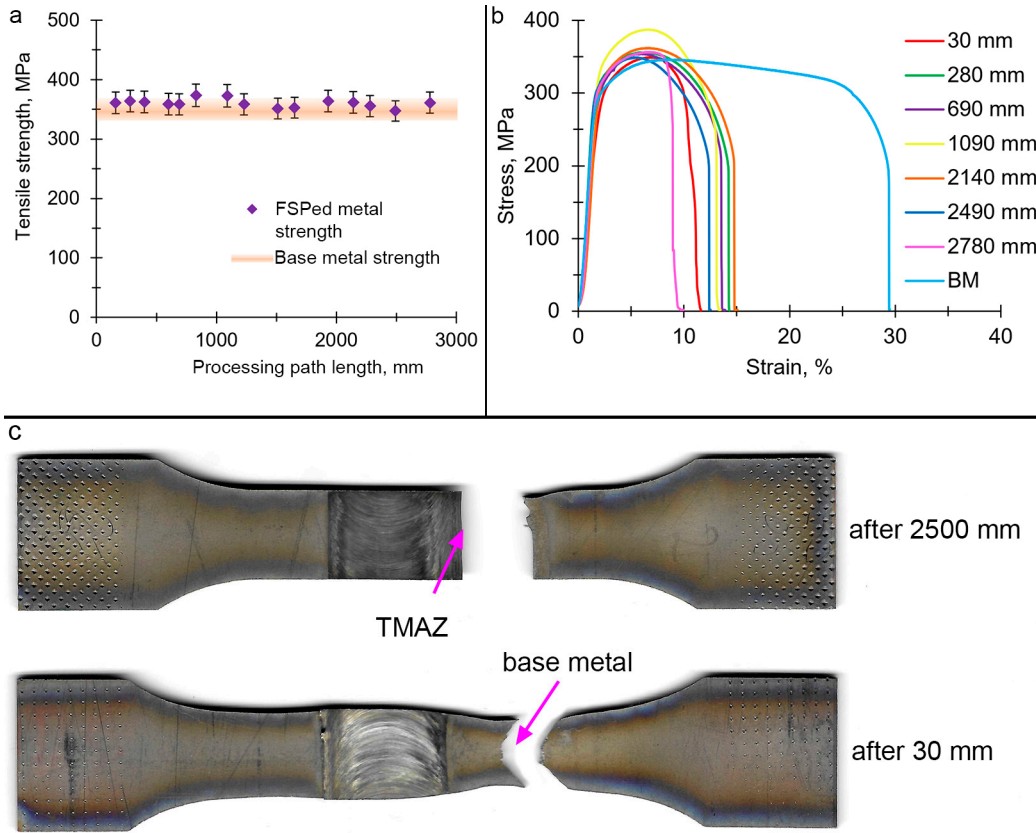

**Figure 21.** Dependence of the weld strength on the tool wear rate (**a**), stress–strain curves of FSPed samples processed by the liquid-cooled ZhS6U tool at different processing path lengths (**b**), and image of samples after fracture (**c**).

### 3.5. Tool Wear Mechanism in Friction Stir Processing of Titanium

The investigation of tool wear during the FSP of titanium revealed an adhesion–diffusion wear mechanism of the tool, as shown schematically in Figure 22. At the beginning of processing, the rotating tool is plunged into the titanium workpiece. The workpiece material is heated up due to friction and goes into a plastic state. Then the tool starts traversing along the weld line. The plasticized titanium is entrained by the tool due to adhesion in the direction from the advancing to the retreating side. This process is accompanied by the formation and growth of a transfer layer on the tool. When the thickness of the layer increases to a critical value and the friction force becomes greater than the adhesion force, the layer peels off partially or completely. The transfer layer diffuses with the tool material at high temperatures, resulting in the formation of a diffusion layer at their interface. The alloy elements of the tool gradually penetrate the transfer layer. At a certain thickness, this layer is detached, together with the transfer layer, and remains in the weld. Thus, the tool gradually wears out, and the surface of the workpiece material is modified by the alloy elements of the tool. The transfer layer is partially preserved on the tool after its removal from the workpiece. Similar effects were found in a paper [41] describing the adhesion of an aluminum alloy to a steel ball with friction. In this pair of materials, the diffusion under these conditions was not so significant. However, the work [42] shows that diffusion layers and intermetallics can also form on the steel tool during the FSW of aluminum.

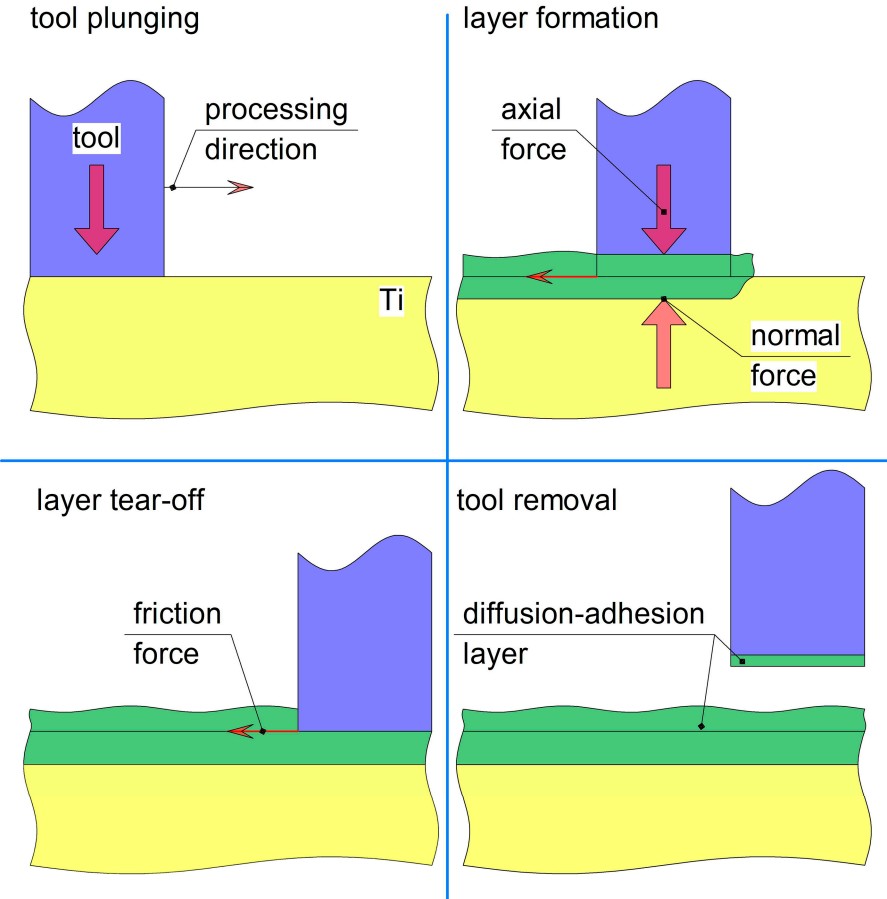

**Figure 22.** Wear mechanism of a Ni-based superalloy tool during FSP of titanium.

### 4. Conclusions

The wear of ZhS32 and ZhS6U nickel superalloy tools was investigated during the friction stir processing of commercially pure titanium with and without liquid cooling, respectively. The effect of tool wear on the quality of the FSP titanium weld was discussed. The total processing path length traversed by the tools without failure was 2790 mm. The liquid-cooled ZhS6U tool was worn much less than the ZhS32 tool, with higher heat resistance. In both cases, the most severe wear was observed at the pin root: the tapered cylindrical pin shape changed to a straight one, and a 1-mm deep depression was formed on the shoulders. The liquid-cooled tool retained its tapered shape much longer. It was also found that the tool overheating during the processing led to the growth of structural elements. In particular, the size of the dendritic colonies, as well as the primary and secondary arm spacings, increased. The results show that the microhardness of the entire ZhS32 tool was 10% lower than that of the as-cast ZhS32 alloy due to the heat effects during the FSP. Liquid cooling allowed the microhardness in the ZhS6U tool to be maintained at the level of the initial material.

The different wear behavior of the tools led to different macrostructural changes in the workpiece material. Changing the tool geometry resulted in changes in the friction conditions and heat generation. The numbers and sizes of the macrodefects were greater in the case of using the ZhS32 tool than when using the liquid-cooled ZhS6U tool. This was because the liquid-cooled tool wore less and retained its original shape for longer. As a result, adhesive mass transfer remained stable for a longer time, contributing to a more stable strength of the processed material. With the ZhS32 tool, the workpiece strength decreased abruptly at a distance of ≈1600 mm due to a sharp increase in the number of defects. With the liquid-cooled ZhS6U tool, no sharp strength drops were observed over the

entire processing interval, i.e., over a span of about 2800 mm. So, the study showed that it is more efficient to apply liquid cooling in friction stir processing for tool life enhancement than to use tools made of more expensive alloys with higher heat resistance. In particular, the lifetime was improved by a factor of 1.75.

The wear of the tool is proposed to occur by an adhesion–diffusion mechanism involving the formation of a diffusion layer between the workpiece and the tool during processing. The detachment of this layer from the tool during mass transfer led to tool wear.

**Author Contributions:** Conceptualization, A.E.; formal analysis, A.A.; investigation, A.A., A.E. and V.B.; methodology, A.E. and V.B.; project administration, A.E.; resources, V.B.; writing—original draft, A.A. and A.E.; writing—review and editing, A.E. All authors have read and agreed to the published version of the manuscript.

**Funding:** The work was performed according to the government research assignment for ISPMS SB RAS, project FWRW–2021–0006. The investigations were carried out using the equipment of the Shared Use Centre Nanotech of ISPMS SB RAS.

**Data Availability Statement:** Not applicable.

**Conflicts of Interest:** The authors declare no conflict of interest.

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
