# Peer review of "Wear of Ni-Based Superalloy Tools in Friction Stir Processing of Commercially Pure Titanium"

_lubricants, doi:10.3390/lubricants11070307_

Round 1

Reviewer 1 Report

Wear of Ni-based superalloy tools in friction stir processing of 2 commercially pure titanium

Line 7: Delete one “correspondence”

Introduction:

Line 33: Why is FSW mentioned first and then FSP? Please explain the differences between the two technologies and refer to papers that justify treating them equally or using the results obtained for FSP as valid for FSW.

Line 53: Please refer to lengths using mm as the unit of measurement. Apply to all the manuscript.

Materials & Methods:

Line 87: Why did the authors decide to analyze pure titanium if in the introduction they said that titanium alloys are applied in aerospace (and many others, adds the reviewer)? Wouldn't it make more sense to study a titanium alloy as a workpiece?

Line 92: The explanation of the cooling system is not very clear. Please explain more about the cooling system used. How can you be sure that the coolant really goes into the channels and the centrifugal force due to rotation does not prevail? Has the coolant flow in the tool been checked? Moreover, what coolant was used? at what temperature?

Line 102: Was the tool tilted?

Line 105: Please refer to lengths using mm as the unit of measurement. Apply to all the manuscript.

Line 110: What were the samples cut with?

Line 114: Up to what grit were the specimens polished?

Line 127: Please specify haw the stretching rate was chosen. Have you followed a standard for the tensile tests?

Line 128: How were the specimens prepared for the Vickers hardness tests?

Line 131: It would be much clearer to insert the referenced figures here to fully describe the methodology and then refer to the methodology to indicate the locations of the Vickers mapping.

Line 132: Why did you choose such a low load? Which standard have you followed? Also the distances between the indentations seem to be too small following the standard for the Vickers hardness measurements.

Results and discussion:

Line 141: How can you be sure that titanium is oxidized? If you do not demonstrate the presence of oxide with an EDX, one can only assume that the titanium has oxidized.

Line 149: It would be useful to see the images and also the numerical results near the end of the machining stroke, i.e. at 3000 mm.

Line 154: On what basis can you compare results obtained with tools having different geometries? And why did you make them different if the goal was their comparison?

Line 155: How did you measured the height of the pin?

Line 155 and line 158: The following two sentences appear to be contradictory:

“The height of the pin did not change”.

“…the pin height increases”.

Line 168: Are you sure it's just due to the cast structure of the billet? What if it was also because of machining to obtain the tools? Have you analyzed a new tool before machining to compare it with the one at the end of machining? It could be useful and interesting.

Line 169: “The tool and the as-cast billet were cut at different angles for structural examination”. Why?

Line 171: “However, this did not interfere with the measurement of the dendritic colony sizes and primary and secondary arm spacings”. On what basis can this statement be made?

Line 177: Please better explain the absence of the specific effect of the process on the dendritic structure.

Table 4: How were the microstructural parameters measured? Were any rules followed? How many measurements were taken to get that standard deviation?

Line 189: Here too it would be very important to compare the properties of the tool before and after machining, not only with the billet as cast.

Line 191: “Large individual deviations can be due to…” and what about some incorrect parameters for the Vickers indentations?

Figure 7: The high dispersions of the measurements near to the 0 are probably due to incorrect distance between the specimen’s edge and incorrect distances between two following indentations. How big are the diagonals of the indent fingerprints? The standard for Vickers micro-hardness requires that between two successive indentations there are between 2 and 5 times the average diagonal measured. Considering such close indentations may lead to misinterpretations influenced by local hardening of the material.

Line 221: How were the areas measured?

Figure 8: it is possible to notice defects even at 5 mm. If there are already flaws, I don't know how much sense it makes to consider tool wear. First you should be sure you get parts free from defects (working on tool geometries or process parameters for example) and then consider tool wear.

Line 243: “Almost free from oxides”. Did the authors perform a EDX analysis to be sure to declare that?

Lines 244-247: Please add some references supporting your statements.

Lines 259-262: I'm not sure it can be said that tensile strength decreases with increasing tool wear. My biggest doubt regarding this research work is that the tensile strength decreases due to the presence of defects (which, as shown, already exist at 5 mm and which therefore cannot be sure of a systematic relationship between tool wear and defects and not between defects and process errors).

Lines 262-264: Please provide the fracture points images.

Line 295: Pleas provide an explanation for this result.

Figure 14: Please be sure that the caption will be in the same page of the relative image.

Lines 317-319: This result leads me to think that the cooling does not actually completely cool the tool.

Line 320: “With other mechanisms.” Which ones? and on what basis is it possible to state it?

Table 5: Same comment of Table 4.

Figure 16: Do you have an EDX image that proves it is deposited titanium?

Figure 17: Oxidized layers and oxidized material have often been mentioned. Where is the oxygen component here in order to be able to demonstrate the possible presence of oxygen?

Line 355: “Rapid cooling”. Perhaps it might be a bit risky to call a cooling rapid, but it hasn't been shown to take effect so far. It seems unlikely to me that those small channels could be able to supply enough liquid to overcome the thermal inertia of the tool. Is there any data that can confirm this hypothesis?

Figure 19: Even here, it is possible to observe some defects in the (a) image. In (b) have you performed EDX analysis to state that that particles were oxides?

Lines 383-385: To make the various tests comparable, the test conditions must always remain constant.

Line 394: Please provide the failure points images.

Paragraph 3.5: Please provide some references or some experimental results for all this section of the manuscript.

Conclusion:

Line 433: This issue is strictly related to the comment of the line 259-262.

Overall comment:

In the introduction it was said that the aim of the work was to study methods for increasing (line 58). I don't think it is the main purpose, which instead is the study of the response in terms of wear of different materials used for the realization of tolls for the FSP.

None

Reviewer 2 Report

Manuscript Number: lubricants-2458519

Title: Wear of Ni-based superalloy tools in friction stir processing of commercially pure titanium

Decision: Major Revision

Article Type: Article

The article is, in general, well written but there are some issues that article should consider to revise in order to improve its quality. Some comments were done in this way:

Ø  The pin tip is given 2 mm. Welded at exactly 2 mm depth. or was it lowered more to increase the friction of the shoulders. This information is not given.

Ø  Let's remove the arrows and texts in Fig 5.

Ø  The hardness change on the pin is attributed to the temperature on the material. Reference should be made to studies that measured the temperature distribution in the source regions. Make use of the following studies.

·       doi.org/10.1515/mt-2022-0277

·       doi.org/10.1016/j.rinp.2019.102814

Ø  Show by point EDX analysis what is shown in Figure 16 as Ti and carbides. The EDX Map shown in Fig 17 does not show this exactly.

Ø  The region shown as oxides in Fig 19-b looks like void. If the authors claim to be oxide, they should prove it with EDX.

Ø  Such voids are formed by the upward transport of the melted material due to its pin shape. Authors can review articles investigating the effect of different pin shapes on weld quality. There are studies in the literature to eliminate these gaps.

Ø  If you give the results numerically and proportionally, the improvement will be easier to understand.

After making the above corrections would recommend this article for publication in Lubricants.

Round 2

Reviewer 1 Report

The comments of the authors are good enough to allow the paper to
be published, in my opinion.

None

Author Response

The author team thanks you for your feedback and evaluation.

Reviewer 2 Report

Manuscript Number: lubricants-2458519

Title: Wear of Ni-based superalloy tools in friction stir processing of commercially pure titanium

Decision: Major Revision

Article Type: Article

The authors made some changes. However, it is not enough. Please complete the revisions below.

Ø  The summary should be revised with the conclusions to be of interest to the readers.

Ø  Introduction section should be expanded. In particular, the temperature factor affecting pin hardness should be taken into account.

Ø  A full photograph of the experimental setup can be attached. (Broader perspective) Thus, important data can be provided for other researchers. Just the 3D view of the Pin is not enough.

Ø  From which part of the material was welding started? Have a hole drilled first as a reference. If so, what is the location and dimensions of this hole? Wait for the pin to heat up and provide sufficient friction. How long is this period? In FSW, this is very important empirical information. Please include this information in the article.

After making the above corrections would recommend this article for publication in Lubricants.

Round 3

Reviewer 2 Report

Accept in present form